# Development of a multiplex Loop-Mediated Isothermal Amplification (LAMP) assay for on-site diagnosis of SARS CoV-2

Woong Sik Jang[1☯], Da Hye Lim[2☯], Jung Yoon[2], Ahran Kim[2], Minsup Lim[1], Jeonghun Nam[1], Richard Yanagihara[3], Sook-Won Ryu[4], Bo Kyeung Jung[5], Nam-Hee Ryoo[6]*, Chae Seung Lim[2]*

1 Emergency Medicine, College of Medicine, Korea University Guro Hospital, Seoul, Republic of Korea, 2 Departments of Laboratory Medicine, College of Medicine, Korea University Guro Hospital, Seoul, Republic of Korea, 3 Pacific Center for Emerging Infectious Diseases Research, John A. Burns School of Medicine, University of Hawaii at Manoa, Honolulu, HI, United States of America, 4 Department of Laboratory Medicine, Kangwon National University, School of Medicine, Kangwondo, Republic of Korea, 5 Department of Laboratory Medicine, Dankook University College of Medicine, Cheonan, Korea, 6 Department of Laboratory Medicine, Dongsan Medical Center, Keimyung University, Daegu, Korea

☯ These authors contributed equally to this work.
* nhryoo@dsmc.or.kr (NHR); malarim@korea.ac.kr (CSL)

**Data Availability Statement:** All relevant data are within the paper and its Supporting Information files.

## Abstract

A newly identified coronavirus, designated as severe acute respiratory syndrome coronavirus 2 (SARS CoV-2), has spread rapidly from its epicenter in China to more than 150 countries across six continents. In this study, we have designed three reverse-transcription loop-mediated isothermal amplification (RT-LAMP) primer sets to detect the RNA-dependent RNA polymerase (RdRP), Envelope (E) and Nucleocapsid protein (N) genes of SARS CoV-2. For one tube reaction, the detection limits for five combination SARS CoV-2 LAMP primer sets (RdRP/E, RdRP/N, E/N, RdRP/E/N and RdRP/N/Internal control (actin beta)) were evaluated with a clinical nasopharyngeal swab sample. Among the five combination, the RdRP/E and RdRP/N/IC multiplex LAMP assays showed low detection limits. The sensitivity and specificity of the RT-LAMP assay were evaluated and compared to that of the widely used Allplex™ 2019-nCoV Assay (Seegene, Inc., Seoul, South Korea) and PowerChek™ 2019-nCoV Real-time PCR kit (Kogenebiotech, Seoul, South Korea) for 130 clinical samples from 91 SARS CoV-2 patients and 162 NP specimens from individuals with (72) and without (90) viral respiratory infections. The multiplex RdRP (FAM)/N (CY5)/IC (Hex) RT-LAMP assay showed comparable sensitivities (RdRP: 93.85%, N: 94.62% and RdRP/N: 96.92%) to that of the Allplex™ 2019-nCoV Assay (100%) and superior to those of PowerChek™ 2019-nCoV Real-time PCR kit (RdRP: 92.31%, E: 93.85% and RdRP/E: 95.38%).

## Introduction

In December 2019, an outbreak in Wuhan, China of a severe respiratory illness was caused by a previously unrecognized coronavirus, which has since been named severe acute respiratory

**Funding:** This study was supported by a government-wide R&D fund project for infectious disease research (HG18C0012), National Research Foundation of Korea (NRF-2016R1A5A1010148), and a grant of the Korea Health Technology R&D Project through the Korea Health Industry Development Institute (KHIDI), funded by the Ministry of Health & Welfare, Republic of Korea (grant number: HR20C0021)

**Competing interests:** No authors have competing interests.

syndrome coronavirus 2 (SARS CoV-2) (genus Betacoronavirus, subgenus Sarbecoronavirus) [1–5]. Clinical signs of the disease, which was subsequently designated coronavirus disease 2019 (COVID-19), included fever, cough and shortness of breath, making it difficult to distinguish from other viral respiratory infections [6,7].

Despite intense efforts to contain the outbreak at the epicenter, the disease has spread throughout China and beyond. The World Health Organization (WHO) declared COVID-19 as a global pandemic, and as of 24 November 2020, Worldometers, which is a real-time international statistics site, announced that the total number of confirmed SARS CoV-2 patients in 220 countries around the world exceeded 59 million, with 1,402,972 that died of the infection. In particular, the USA and India have respectively accumulated over 12 million (263,687 deaths) and 9 million cases (134,254 deaths) of SARS CoV-2. SARS CoV-2 is still spreading worldwide, and there is an urgent need to conduct rapid diagnosis followed by patient isolation and treatment. Currently, an RT-qPCR-based test distributed by WHO is being deployed in many countries to detect SARS CoV-2 RNA, and several commercial RT-qPCR kits (PowerChek™ 2019-nCoV Real-time PCR kit [Kogenebiotech, Seoul, South Korea], granted EUAL in Korea) are available to diagnose SARS CoV-2 in Korea. However, these RT-qPCR detection methods require nearly three hours to produce results, and skilled technicians and advanced laboratory infrastructure are necessary. As a result, testing is limited to institutions in which specialized medical services are available and in areas where wide-scale surveillance is required.

Loop-mediated isothermal amplification (LAMP) is a highly sensitive, low-cost, single-tube technology to detect the target nucleic acid sequences [2,8]. Typically, six primers, including four primers selected by combining parts of the target DNA and two additional loop primers, are used to amplify a specific gene region. Bst DNA polymerase, a strand-displacement DNA polymerase, enables a loop structure formation for the inner primers, producing LAMP's unique rapid self-priming amplification [9,10]. LAMP has been widely applied to detect various microbial pathogens [11–13]. In particular, reverse-transcription LAMP (RT-LAMP) has been used for point-of-care-testing for RNA virus infections [14].

In this study, we have developed multiplex SARS CoV-2 LAMP primer/probe sets using strand-displaceable probes, based on the region of the RdRP, E and N gene of the aligned sequences of SARS CoV-2 subtypes. Among five combination SARS CoV-2 LAMP primer sets, RdRP/N/internal control (actin beta, IC) multiplex LAMP assay showed the lowest detection limits. The performance of the multiplex SARS CoV-2 RdRP/N/IC LAMP assay was compared with direct RT-qPCR methods using the Seegene Allplex™ 2019-nCoV Assay and Kogenebiotech PowerChek™ 2019-nCoV Real-time PCR kit for SARS CoV-2 clinical samples.

## Materials and methods

### Clinical samples and RNA extraction

This study was approved by the Medical Ethics Committee of Korea University's Guro Hospital (2019GR0055). Informed consent was waived by the Institutional Review Board (IRB) because this study used residual samples. To estimate the number of samples required for clinical test of the multiplex RT-LAMP assay, the following formula was used:

$$n \geq \frac{(1.96)^2 p(1-p)}{x^2}$$

where $p$ is the suspected sensitivity, and $x$ is the desired margin of error [15,16]. The true-positive rate (sensitivity) was defined as the proportion of SARS-CoV-2 positive which is correctly identified by the multiplex RT-LAMP assay compared to the Allplex™

2019-nCoV Assay (Seegene, Inc., Seoul, South Korea). We suspected the sensitivity and specificity of the multiplex RT-LAMP assay to be 95% with a desired margin of error of 0.04%. Under these conditions, the number of required samples is 114.0475 (rounded up to 115) per group. In this experiment, we have tested total 292 samples (130 positive and 162 negative). A total of 130 clinical samples, including nasopharyngeal (NP) swabs, oropharyngeal (OP) swabs, sputum, saliva and urine, were collected from 91 patients suspected of being infected with SARS CoV-2 in the Republic of Korea. All clinical samples were confirmed using the Allplex[TM] 2019-nCoV Assay (Seegene, Inc., Seoul, South Korea) and PowerChek™ 2019-nCoV Real-time PCR kit (Kogenebiotech, Seoul, South Korea). To assess the specificity of the multiplex SARS CoV-2 RT-LAMP assay, 162 NP swab specimens were tested from individuals with (72) and without (90) viral respiratory infections. Respiratory viral infections, as confirmed by PCR using the Anyplex[TM] II RV16 detection kit, included 39 coronavirus (KHU1, NL63, 229E), 3 influenza virus A/H1N1, 3 influenza virus A/H3N2, 3 influenza virus B, 3 respiratory syncytial virus (RSV) A, 3 RSV B, 3 adenovirus, 3 parainfluenzavirus (PIV) types 1 to 4, 3 human bocavirus (HboV), 3 human enterovirus (HEV), 3 human rhinovirus (HRV) and 3 metapneumovirus (MPV). RNA was extracted from 200 μL of SARS CoV-2 clinical samples using an InviMag Universal RNA Mini Kit (Stratec Molecular, Berlin, Germany), according to the manufacturer's manual. RNA extraction from the 162 NP swab controls was performed using the QIAamp Viral RNA Mini kit (Qiagen, Hilden, Germany), according to the manufacturer's instructions. RNA was stored at -50˚C. The SARS CoV-2 RT-LAMP was performed blindly with the operator unaware of any previous test results.

## Primer design

The RT-LAMP primer sets for SARS CoV-2 were designed from conserved regions of the RdRP, E and N genes (Table 1). All LAMP primers including two outer primers (forward primer F3 and backward primer B3), two inner primers (forward inner primer FIP and backward inner primer BIP), and two loop primers (forward loop primer LF and backward loop primer LB) were designed using the Primer Explorer version 4 software (Eiken Chemical Co., Tokyo, Japan). A 32-oligomer or 35-oligomer fluorophore strand-displaceable probes was designed at the 5' end of the LB primer, and the 30-oligonucleotide or 35-oligonucleotide quencher was complementary to the probe. Strand-displaceable probes were 5′-labeled with FAM, Hex and Cy5 for RdRP, E and N, respectively. Before use in LAMP, all primers were assessed for specificity by performing a BLAST search. All LAMP primers and probes were synthesized by Macrogen, Inc (Seoul, South Korea).

## Real-time RT-PCR

To evaluate the performance of the multiplex SARS CoV-2 RT-LAMP assay, two real-time RT-PCR tests, using the PowerChek™ 2019-nCoV Real-time PCR kit (Kogenebiotech, Seoul, Korea) and Allplex[TM] 2019-nCoV Assay (Seegene, Inc., Seoul, South Korea), were performed by using CFX96 Touch Real time PCR detection System (Bio-Rad, USA). For the PowerChek™ 2019-nCoV Real-time PCR kit, the thermocycling parameters were used as follows: reverse transcription at 50˚C for 30 min, inactivation at 95˚C for 10 min, 40 cycles of denaturation at 95˚C for 15 s, and annealing with fluorescence detection at 60˚C for 1 min. The PCR cycling conditions of the Allplex[TM] 2019-nCoV Assay were as follows: reverse transcription at 50˚C for 20 min, inactivation at 95˚C for 15 min, 45 cycles of denaturation at 95˚C for 15 s, and annealing with fluorescence detection at 58˚C for 30 sec.

**Table 1. The multiplex SARS CoV-2 RT-LAMP primer sets used in this study.**

| Target | Name | Sequence (5'-3') | Length (mer) |
|---|---|---|---|
| SARS CoV-2 (RdRP gene) | RdRP F3 | CGA TAA GTA TGT CCG CAA TT | 20 |
| | RdRP B3 | GCT TCA GAC ATA AAA ACA TTG T | 22 |
| | RdRP FIP | ATG CGT AAA ACT CAT TCA CAA AGT CCA ACA CAG ACT TTA TGA GTG TC | 47 |
| | RdRP BIP | TGA TAC TCT CTG ACG ATG CTG TTT AAA GTT CTT TAT GCT AGC CAC | 45 |
| | RdRP BLP | TCA ATA GCA CTT ATG CAT CTC AAG G | 25 |
| | RdRP FLP | TGT GTC AAC ATC TCT ATT TCT ATA G | 25 |
| | RdRP BLP probe 1 | [FAM]-CGG GCC CGT ACA AAG GGA ACA CCC ACA CTC CGT CAA TAG CAC TTA TGC ATC TCA AGG | 57 |
| SARS CoV-2 (E gene) | E F3 | TCA TTC GTT TCG GAA GAG A | 19 |
| | E B3 | AGG AAC TCT AGA AGA ATT CAG AT | 23 |
| | E FIP | TGT AAC TAG CAA GAA TAC CAC GAA ACA GGT ACG TTA ATA GTT AAT AGC G | 49 |
| | E BIP | GCT TCG ATT GTG TGC GTA CTC GAG AGT AAA CGT AAA AAG AAG G | 43 |
| | E BLP | GCT GCA ATA TTG TTA ACG TGA GTC | 24 |
| | E BLP probe 1 | [Hex]-CGG GCC CGT ACA AAG GGA ACA CCC ACA CTC CGG CTG CAA TAT TGT TAA CGT GAG TC | 56 |
| SARS CoV-2 (N gene) | N F3 | TGG ACC CCA AAA TCA GCG | 18 |
| | N B3 | GCC TTG TCC TCG AGG GAA T | 19 |
| | N FIP | CCA CTG CGT TCT CCA TTC TGG TAA ATG CAC CCC GCA TTA CG | 41 |
| | N BIP | CGC GAT CAA AAC AAC GTC GGC CCT TGC CAT GTT GAG TGA GA | 41 |
| | N BLP | GGT TTA CCC AAT AAT ACT GCG TCT T | 25 |
| | N FLP | TGA ATC TGA GGG TCC ACC AAA | 21 |
| | N BLP probe 2 | [Cy5]-GTC AGT GCA GGC TCC CGT GTT AGG ACG AGG GTA GGG GTT TAC CCA ATA ATA CTG CGT CTT | 60 |
| Human (actin beta gene) | IC F3 | AGT ACC CCA TCG AGC ACG | 18 |
| | IC B3 | AGC CTG GAT AGC AAC GTA CA | 20 |
| | IC FIP | GAG CCA CAC GCA GCT CAT TGT ATC ACC AAC TGG GAC GAC A | 40 |
| | IC BIP | CTG AAC CCC AAG GCC AAC CGG CTG GGG TGT TGA AGG TC | 38 |
| | IC BLP | CGA GAA GAT GAC CCA GAT CAT GT | 23 |
| | IC FLP | TGT GGT GCC AGA TTT TCT CCA | 21 |
| | IC BLP probe 1 | [HEX]–CGG GCC CGT ACA AAG GGA ACA CCC ACA CTC CGC GAG AAG ATG ACC CAG ATC ATG T | 55 |
| Quencher probe 1 | | GAG TGT GGG TGT TCC CTT TGT ACG GGC CCG -BHQ1 | 30 |
| Quencher probe 2 | | CCT ACC CTC GTC CTA ACA CGG GAG CCT GCA CTG AC -BHQ2 | 35 |

## Multiplex RT-LAMP

The RT-LAMP assay was performed with the Miso® RNA amplification kit (Mmonitor, Daegu, South Korea). For multiplex SARS CoV-2 RdRP/N/IC RT-LAMP assay, the reaction mixture was prepared with 12.5 μL of 2x reaction buffer, 1.2 μL of SARS CoV-2 RdRP gene LAMP primer mix, 0.6 μL of SARS CoV-2 N gene LAMP primer mix, 0.6 μL of internal control LAMP primer mix, 600 nM quencher 1 solution, 240 nM quencher 2 solution, 2 μL of

enzyme mix, and 2.5 μL of sample RNA (final reaction volume 25 μL). The compositions of all LAMP primer mix were 4 μM of two outer primers (F3 and B3) and 3.32 μM of two inner primers (FIP and BIP), 10 μM of loop LF primer, 4 μM loop LB primer, and 6 μM loop LB probe primer. The RT-LAMP assay was run on CFX 96 Touch Real-Time PCR Detection System (Bio-Rad Laboratories, Hercules, CA, USA) at 60˚C for 40 min. The FAM, Hex and Cy5 fluorescence channels were used for detecting RdRP, E and N gene, respectively.

### Limits of detection

pTOP Blunt V2 plasmids, including partial RdRP, E or N gene sequences of SARS CoV-2, were used to test the limit of detection of the RT-LAMP assay. All plasmids were constructed by Macrogen, Inc. (Seoul, South Korea). The plasmids were serially diluted 10-fold from $1 \times 10^8$ copies/μL to $1 \times 10^0$ copies/μL to determine the detection of limit of the multiplex SARS CoV-2 RdRP/N/IC RT-LAMP assay. In addition, the detection limit of the multiplex SARS CoV-2 RdRP/N/IC RT-LAMP was tested on 10-fold serially diluted clinical samples from SARS CoV-2 patients.

## Results

### Optimization of the multiplex SARS CoV-2 RT-LAMP assay

The sensitivity of the SARS CoV-2 RdRP, E and N gene RT-LAMP was evaluated by testing synthetic plasmid standards, including synthetic partial RdRP, E and N genes ranging from $10^8$ to $10^0$ copies/μL, respectively (Fig 1). The limits of detection for the RdRP gene E gene and N gene were $1x10^1$ copies/μL, $1x10^1$ copies/μL and $1x10^2$ copies/μL, respectively. For multiplex SARS CoV-2 RT-LAMP in one tube, four combination of RdRP (FAM)/E (Hex), RdRP (FAM)/N (Cy5), E (Hex)/N (Cy5) and RdRP (FAM)/E (Hex)/N (Cy5) were tested using strand-displaceable probes. For optimization of four multiplex SARS CoV-2 RdRP/E, RdRP/N, E/N and RdRP/E/N LAMP assays, different ratios (1:1, 1:0.5, 1:1.5 or 1:1:1, 0.8:1:0.5, 1:1:0.5) of primers for the RdRP/E, RdRP/N, E/N and RdRP/E/N were tested, using synthetic RdRP, E and N gene plasmids (Table 2). Among the three ratios of both the RdRP/N primer set and E/N primer set, a ratio of 1:0.5 showed faster Ct values (12.64/12.03 and 14.04/11.6, respectively) and the most stable graph. In the case of the RdRP/E primer set and RdRP/E/N primer set, the ratio of 1:1 and 0.8:1:0.5 showed faster Ct values (13.41/11.07 and 14.48/17.22/15.99, respectively). Next, temperature-gradient tests (60, 62 and 65˚C) showed that the optimum temperature was 60˚C (Table 3), which is early Ct values of all four combination LAMP primer sets (RdRP/E: 13.41/11.07, RdRP/N: 12.77/12.85, E/N: 14.04/11.6, RdRP/E/N: 15.97/15.14/13.67).

### Comparison of detection limits of the multiplex SARS CoV-2 RT-LAMP assay with two commercial RT-qPCR assays for SARS CoV-2 clinical samples

The detection limits of monoplex SARS CoV-2 LAMP primer sets were compared to those of two commercial RT-qPCR kits (Allplex™ 2019-nCoV Assay and PowerChek™ 2019-nCoV Real-time PCR kit) for 10-fold serial dilutions of SARS CoV-2 NP samples (range of $10^{-3}$–$10^{-7}$) (Table 4 and Fig 2). Monoplex RdRP, E and N RT-LAMP primer sets showed detection limits of $10^{-5}$, $10^{-5}$ and $10^{-6}$. The combination of RdRP/E and RdRP/N primer sets showed detection limits of $10^{-5}/10^{-5}$ and $10^{-5}/10^{-6}$, respectively, whereas combination of E/N and RdRP/E/N primer sets showed detection limits of $10^{-3}/10^{-4}$ and $10^{-3}/10^{-3}/10^{-4}$, respectively. In addition, the multiplex SARS CoV-2 RdRP (FAM)/N (Cy5)/ internal control (IC, actin beta, HEX) RT-LAMP assay is also developed to confirm the success of the extraction step.

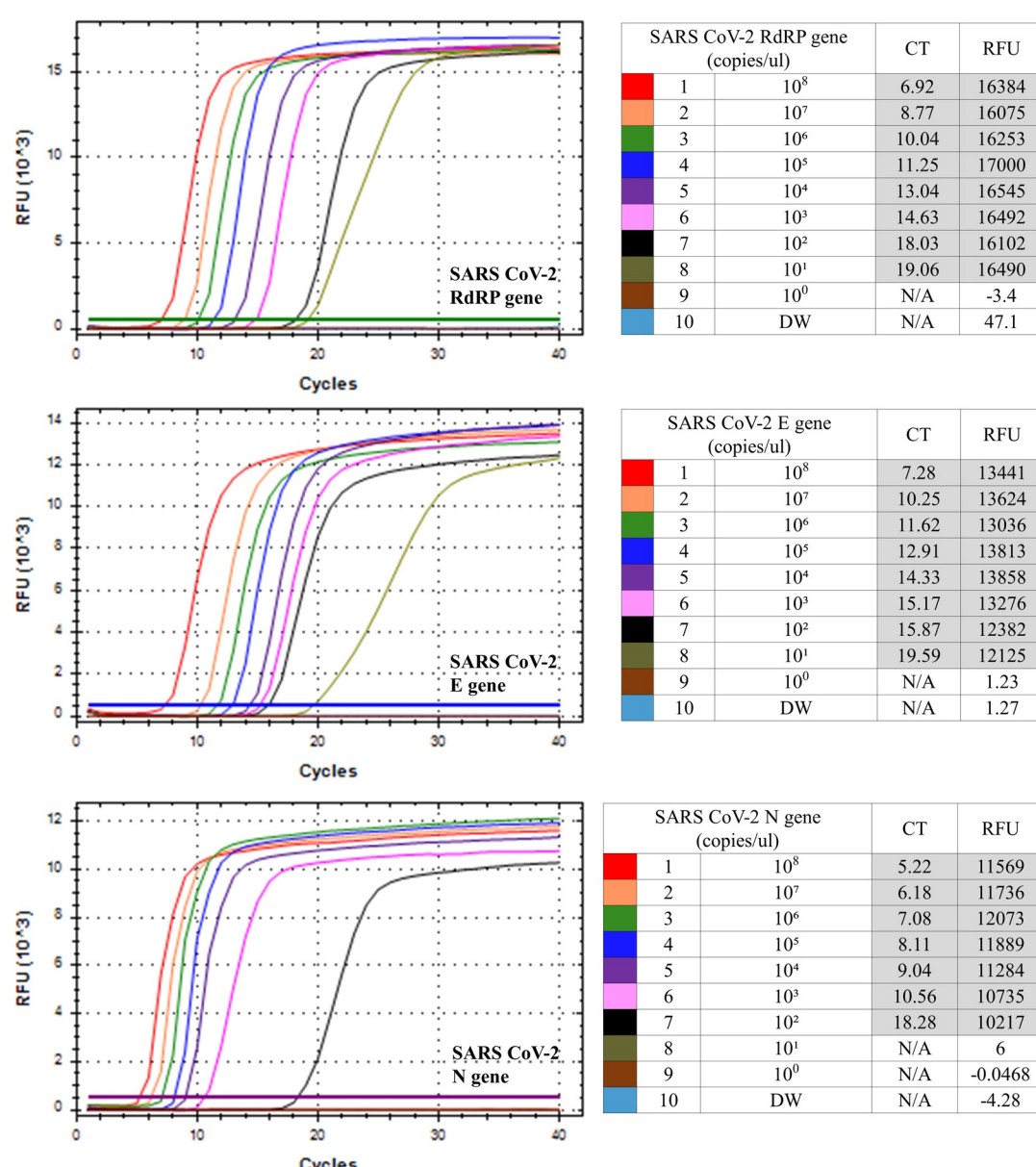

**Fig 1. Limit of detection for the monoplex SARS CoV-2 RT-LAMP assay.** The limit of detection for the monoplex SARS CoV-2 RdRP (A), E (B) and N (C) RT-LAMP assay was tested with synthetic RdRP, E and N plasmid ranging from $10^8$ to $10^0$ copies/µL, respectively. Numbers (1–10) indicated plasmid copy numbers/µL ($1.0 \times 10^8$–$1.0 \times 10^0$ copies/µL) and negative control (distilled water (DW) as non-template control).

Adding internal control (actin ß) LAMP primer set to RdRP and N (ratio of RdRP/N/ IC = 1:0.5:0.5) LAMP assay did not affect the detection limits of RdRP and N LAMP assay. The SARS CoV-2 RdRP (FAM)/N (Cy5)/ IC (Hex) RT-LAMP assay showed the detection limit of $10^{-5}/10^{-6}$ for RdRP and N (Table 4 and Fig 2C) but signal of IC was not detected. Although the signal of IC in SARS CoV-2 RdRP (FAM)/N (Cy5)/ IC (Hex) RT-LAMP is unstable in SARS CoV-2 clinical samples, signal of IC in the RT-LAMP assay was specifically detected until $10^{-2}$ diluted samples for non-infected clinical samples (S1 Fig). Among five combination LAMP primer sets, the multiplex SARS CoV-2 RdRP/N/IC RT-LAMP assay

**Table 2. Different concentration ratios of the SARS CoV-2 RdRP/E, RdRP/N, E/N and RdRP/E/N primer sets (1:1, 1:0.5 and 1:1.5, respectively) for the synthetic SARS CoV-2 RdRP, E and N gene plasmids.**

| | ratios of primers | | 1:1 (1:1:1) | | 1:0.5 (0.8:1:0.5) | | 1:1.5 (1:1:0.5) | |
|---|---|---|---|---|---|---|---|---|
| | Primer set | Plasmid (10$^7$) | CT | RFU | CT | RFU | CT | RFU |
| Monoplex RT-LAMP | RdRP (FAM) | RdRP | **10.73** | **31987** | | | | |
| | E (Hex) | E | **11.11** | **10860** | | | | |
| | N (Cy5) | N | **7.51** | **10689** | | | | |
| Multiplex RT-LAMP | RdRP (FAM) + E (Hex) | RdRP | **13.41** | **9145** | 24.16 | 9333 | 27.09 | 603 |
| | | E | **11.07** | **3950** | 28.06 | 888 | 21.96 | 2547 |
| | RdRP (FAM) + N (Cy5) | RdRP | 19.05 | 18554 | **12.64** | **22613** | 35.74 | 14778 |
| | | N | 7.23 | 10383 | **12.03** | **4803** | 6.12 | 16188 |
| | E (Hex) + N (Cy5) | E | 29.21 | 10780 | **14.04** | **14255** | 54.28 | 1484 |
| | | N | 7.39 | 8410 | **11.6** | **3775** | 6.76 | 11479 |
| | RdRP (FAM) + E (Hex) + N (Cy5) | RdRP | 13.84 | 16578 | **14.48** | **14187** | 17.53 | 11088 |
| | | E | 35.13 | 1546 | **17.22** | **7820** | 18.51 | 8421 |
| | | N | 9.39 | 7877 | **15.99** | **2907** | 21.19 | 2038 |

showed the lowest detection limits. Furthermore, the multiplex SARS CoV-2 RdRP/N/IC RT-LAMP assay showed comparable sensitivities with those of the Allplex™ 2019-nCoV Assay and lower detection limits than those of the PowerChek™ 2019-nCoV Real-time PCR kit. Thus, the multiplex SARS CoV-2 RdRP/N/IC RT-LAMP assays were further tested with SARS CoV-2 clinical samples.

## Comparison of the clinical performance of the multiplex SARS CoV-2 RdRP/N/IC RT-LAMP assay with that of Allplex™ 2019-nCoV Assay, and PowerChek™ 2019-nCoV Real-time PCR kit using clinical samples

To confirm the clinical performance of the multiplex SARS CoV-2 RdRP/E and RdRP/N/IC RT-LAMP, the sensitivities and specificities of the assays were compared to those of the Allplex™ 2019-nCoV Assay, and PowerChek™ 2019-nCoV Real-time PCR kit for 130 clinical samples from 91 SARS CoV-2 patients and 162 NP specimens from individuals with (72) and without (90) viral respiratory infections (Table 5). For the SARS CoV-2 clinical samples

**Table 3. Temperature gradient tests (60, 62 and 65°C) of the four multiplex combination RT-LAMP assay.**

| Temperature (°C) | | | 60°C | | 62°C | | 65°C | |
|---|---|---|---|---|---|---|---|---|
| | Primer set | Plasmid (10$^7$) | CT | RFU | CT | RFU | CT | RFU |
| Monoplex RT-LAMP | RdRP (FAM) | RdRP | **11.01** | **31987** | 11.09 | 27937 | 15.94 | 21763 |
| | E (Hex) | E | **11.11** | **10860** | 13.65 | 10250 | 31.87 | 7281 |
| | N (Cy5) | N | **7.51** | **10689** | 7.53 | 10238 | 9.61 | 7086 |
| Multiplex RT-LAMP | RdRP (FAM) + E (Hex) | RdRP | **13.41** | **9145** | 13.06 | 10589 | 22.01 | 7694 |
| | | E | **11.07** | **3950** | 16.55 | 1388 | 25.64 | 949 |
| | RdRP (FAM) + N (Cy5) | RdRP | **12.77** | **22613** | 12.24 | 20545 | 16.42 | 15942 |
| | | N | **12.85** | **4803** | 13.28 | 4442 | 14.88 | 4201 |
| | E (Hex) + N (Cy5) | E | **14.04** | **14255** | 15.32 | 14261 | Neg | 324 |
| | | N | **11.6** | **3775** | 11.78 | 4411 | 14.44 | 3869 |
| | RdRP (FAM) + E (Hex) + N (Cy5) | RdRP | **15.97** | **14608** | 15.12 | 13649 | 25.2 | 7986 |
| | | E | **15.14** | **11522** | 18.75 | 6030 | Neg | 687 |
| | | N | **13.67** | **3757** | 13.85 | 3791 | 17.93 | 3504 |

**Table 4. Limit of detection (LOD) tests of the monoplex and multiplex RT-LAMP assay, Allplex™ 2019-nCoV Assay and PowerChek™ 2019-nCoV Real-time PCR kit for clinical SARS CoV-2 NP sample (range of $10^{-3}$–$10^{-7}$).**

| | dilution | | nasopharyngeal swab clinical sample | | | | | | | | | |
| --- | --- | --- | --- | --- | --- | --- | --- | --- | --- | --- | --- | --- |
| | | | $10^{-3}$ | | $10^{-4}$ | | $10^{-5}$ | | $10^{-6}$ | | $10^{-7}$ | |
| | Primer set | | CT | RFU | CT | RFU | CT | RFU | CT | RFU | CT | RFU |
| Monoplex RT-LAMP | RdRP (FAM) | RdRP | 15.15 | 8685 | 18.02 | 8580 | 21.03 | 8305 | Neg | - | Neg | - |
| | E (Hex) | E | 12.24 | 15456 | 14.05 | 15202 | 17.22 | 15812 | Neg | - | Neg | - |
| | N (Cy5) | N | 14.87 | 5023 | 17.08 | 4938 | 19.7 | 4725 | 22.56 | 4963 | Neg | - |
| Multiplex RT-LAMP | RdRP (FAM) + E (Hex) | RdRP | 14.23 | 7180 | 15.47 | 7586 | 17.49 | 7680 | Neg | - | Neg | - |
| | | E | 15.25 | 4997 | 17.56 | 5372 | 20.25 | 4535 | Neg | - | Neg | - |
| | RdRP (FAM) + N (Cy5) | RdRP | 15.18 | 7967 | 18.19 | 8148 | 25.18 | 7567 | Neg | - | Neg | - |
| | | N | 13.25 | 5196 | 15.05 | 5344 | 17.2 | 5687 | 19.77 | 6746 | Neg | - |
| | E (Hex) + N (Cy5) | E | 24.84 | 14481 | Neg | - | Neg | - | Neg | - | Neg | - |
| | | N | 16.31 | 14517 | 25.2 | 15850 | Neg | - | Neg | - | Neg | - |
| | RdRP (FAM) + E (Hex) + N (Cy5) | RdRP | 25.9 | 6840 | Neg | - | Neg | - | Neg | - | Neg | - |
| | | E | 30.61 | 2478 | Neg | - | Neg | - | Neg | - | Neg | - |
| | | N | 22.01 | 3825 | 25.19 | 4233 | Neg | - | Neg | - | Neg | - |
| | **RdRP (FAM) + N (Cy5) + Internal control (Hex)** | RdRP | 17.56 | 6555 | 23.33 | 6350 | 27.84 | 5890 | Neg | - | Neg | - |
| | | N | 17.22 | 4444 | 19.92 | 4750 | 23.46 | 4753 | 29.03 | 4986 | Neg | - |
| | | IC | Neg | - | Neg | - | Neg | - | Neg | - | Neg | - |
| Allplex™ 2019-nCoV Assay | E (FAM) + RdRP (Texas red) + N (Cy5) | E | 27.74 | 4639 | 32.05 | 4421 | 37.85 | 2148 | 38.88 | 1710 | Neg | - |
| | | RdRP | 29.47 | 5824 | 33.73 | 4631 | 39.06 | 1964 | Neg | - | Neg | - |
| | | N | 29.59 | 2982 | 33.79 | 2805 | 38.46 | 2115 | Neg | - | Neg | - |
| PowerChek™ 2019-nCoV Real-time PCR kit | RdRP (FAM)/E (FAM) | RdRP | 27.09 | 14627 | 31.85 | 9908 | Neg | - | Neg | - | Neg | - |
| | | E | 28.51 | 14404 | 33.26 | 6377 | Neg | - | Neg | - | Neg | - |

(n = 130), the sensitivities of the Allplex™ 2019-nCoV Assay for RdRP, E, and N gene were all 100%, excepted for Internal control (IC, 97.69%) and those of PowerChek™ 2019-nCoV Real-time PCR kit for RdRP, E and RdRP/E were 92.31%, 93.85% and 95.38%, respectively. The sensitivities of the multiplex SARS CoV-2 RdRP/N/IC RT-LAMP were 93.85% in the RdRP channel (FAM), 94.62% in the N channel (Cy5), 50.77% in the internal control channel (HEX) and 96.92% in RdRP or N channels. The specificities of two assays for SARS CoV-2 negative clinical samples (n = 162) were 100%, excepted for PowerChek™ 2019-nCoV Real-time PCR kit (99.38%) (Table 5). The sensitivity of the internal control channel of the multiplex SARS CoV-2 RdRP/N/IC RT-LAMP assay, Allplex™ 2019-nCoV Assay, and PowerChek™ 2019-nCoV Real-time PCR kit for SARS CoV-2 negative clinical samples was 100%. Overall, the sensitivity for SARS CoV-2 clinical samples were the highest in the test of the Allplex™ 2019-nCoV Assay, followed by the multiplex SARS CoV-2 RdRP/N/IC RT-LAMP assay and finally the Power-Chek™ 2019-nCoV Real-time PCR kit.

## Cross-reactivity tests of the multiplex SARS CoV-2 RdRP/N/IC RT-LAMP, Allplex™ 2019-nCoV Assay and PowerChek™ 2019-nCoV Real-time PCR kit with other respiratory viruses

To confirm the absence of cross-reactivity with other common respiratory viruses, NP swabs from 72 patients with known infections with 39 Coronavirus (229E, NL63 and OC43), 6 influenza virus A/ B, 6 RSV A/ B, 3 adenovirus, 3 PIV, 3 HBoV, 3 HEV, 3 HRV and 3 MPV were tested by the multiplex SARS CoV-2 RdRP/N/IC RT-LAMP assay, Allplex™ 2019-nCoV Assay (Seegene, Inc., Seoul, South Korea) and PowerChek™ 2019-nCoV Real-time PCR kit

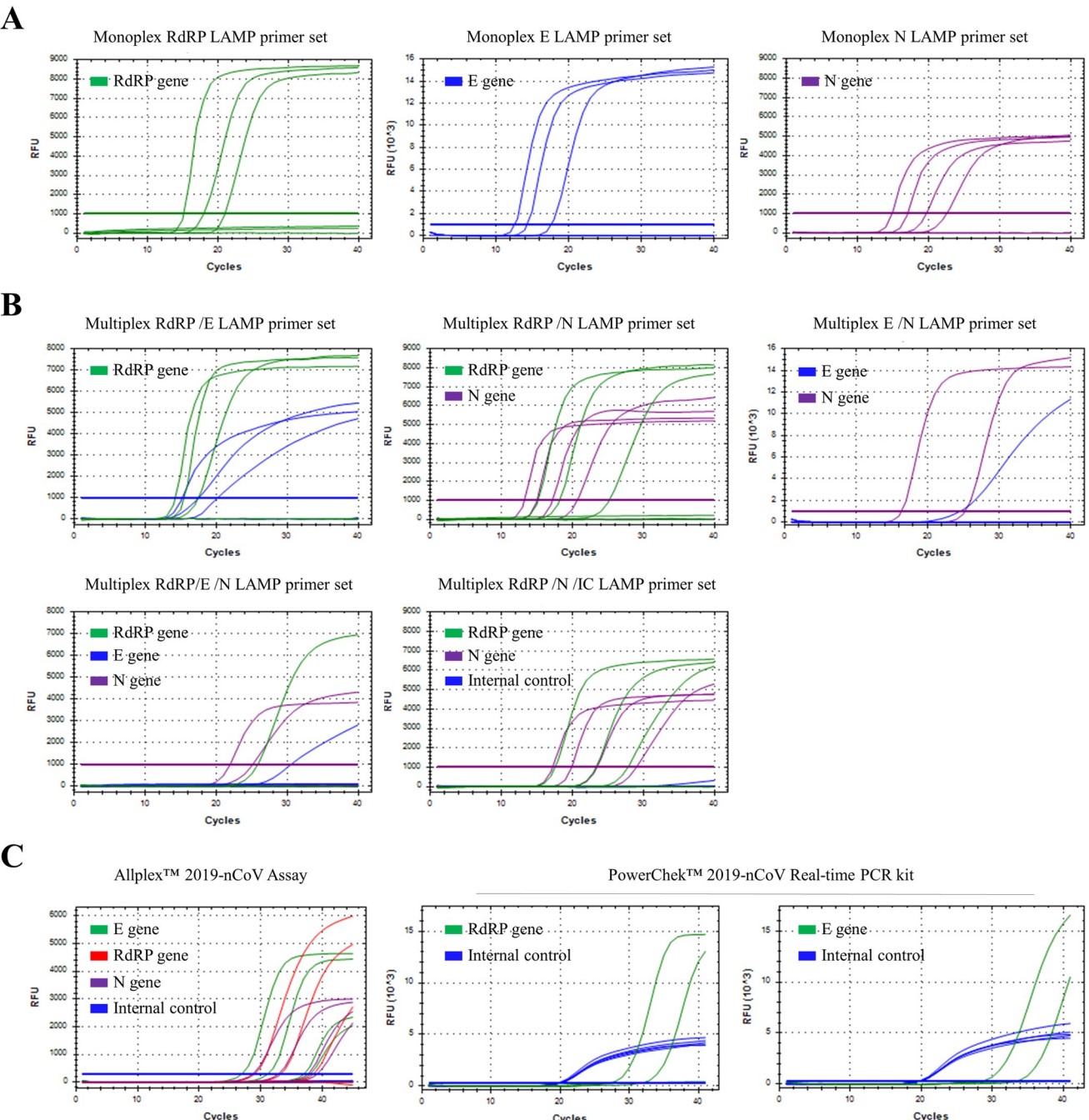

**Fig 2. Limit of detection (LOD) tests of the monoplex and multiplex RT-LAMP assay, Allplex™ 2019-nCoV Assay and PowerChek™ 2019-nCoV Real-time PCR kit for clinical SARS CoV-2 NP sample (range of $10^{-3}$–$10^{-7}$).** (A) Detection limits of monoplex SARS CoV-2 RdRP (left), E (middle) and N (right) LAMP primer sets. (B) Detection limits of the multiplex SARS CoV-2 RdRP/E (upper left), RdRP/N (upper middle), E/N (upper right), RdRP/E/N (lower left) and RdRP/N/IC (lower middle) LAMP primer sets (C) Detection limits of Allplex™ 2019-nCoV assay (Left) and PowerChek™ 2019-nCoV Real-time PCR kit (Right).

(Kogenebiotech, Seoul, Korea) (Table 6). As a result, all three molecular diagnostic tests showed no cross-reactivity with other infectious viruses. Particularly, the multiplex SARS CoV-2 RdRP/N/IC RT-LAMP assay do not cross-react with human coronavirus 229E, NL63 and OC43.

**Table 5. Comparison of clinical performance of the multiplex SARS CoV-2 RT-LAMP assay with Allplex™ 2019-nCoV Assay and Powerchek™ 2019-nCoV Real-time PCR kit for SARS CoV-2 in clinical samples.**

| Clinical samples | | 2019-nCoV (n = 130) | | Non-infection (n = 162) | | |
|---|---|---|---|---|---|---|
| | | P/N | Sensitivity | P/N | Sensitivity | Specificity |
| Allplex™ 2019-nCoV Assay | E (FAM) | 130/0 | **100%** | 0/162 | | **100%** |
| | RdRP (Texas red) | 130/0 | **100%** | 0/162 | | **100%** |
| | N (Cy5) | 130/0 | **100%** | 0/162 | | **100%** |
| | IC (Hex) | 127/3 | **97.69%** | 162/0 | **100%** | |
| | E/RdRP/N | 130/0 | **100%** | 0/162 | | **100%** |
| SARS CoV-2 RdRP/N/IC RT-LAMP assay | RdRP (FAM) | 122/8 | **93.85%** | 0/162 | | **100%** |
| | N (Cy5) | 123/7 | **94.62%** | 0/162 | - | **100%** |
| | IC (Hex) | 66/64 | **50.77%** | 162/0 | **100%** | |
| | RdRP/N | 126/4 | **96.92%** | 0/162 | - | **100%** |
| Powerchek™ 2019-nCoV Real-time PCR kit | RdRP (FAM) | 120/10 | **92.31%** | 0/162 | - | **100%** |
| | IC (Hex) | 130/0 | **100%** | 162/0 | **100%** | - |
| | E (FAM) | 122/8 | **93.85%** | 1/161 | - | **99.38%** |
| | IC (Hex) | 130/0 | **100%** | 162/0 | **100%** | - |
| | RdRP/E | 124/6 | **95.38%** | 1/161 | | **99.38%** |

Sensitivity and specificity were calculated by taking Allplex™ 2019-nCoV Assay as standard. P/N: Positive/negative ratio.

## Discussion

SARS CoV-2 (family *Coronaviridae*, genus *Betacoronavirus*) is a positive-sense, single-stranded RNA virus [17], and it represents the seventh coronavirus known to infect humans, the others being 229E, NL63, OC43, HKU1, Middle East respiratory syndrome coronavirus

**Table 6. Cross-reactivity of the multiplex SARS CoV-2 RdRP/N/IC RT-LAMP assay with Allplex™ 2019-nCoV Assay and Powerchek™ 2019-nCoV Real-time PCR kit for SARS CoV-2 against other human infectious viruses.**

| Virus | No | SARS CoV-2 RdRP/N/IC RT-LAMP assay | | Allplex™ 2019-nCoV Assay | | | Powerchek™ 2019-nCoV Real-time PCR kit | |
|---|---|---|---|---|---|---|---|---|
| | | RdRP (FAM) | N (Cy5) | E (FAM) | RdRP (Texas red) | N (Cy5) | RdRP (FAM) | E (FAM) |
| **CoV 229E** | 13 | 0/13 | 0/13 | 0/13 | 0/13 | 0/13 | 0/13 | 0/13 |
| **CoV NL63** | 13 | 0/13 | 0/13 | 0/13 | 0/13 | 0/13 | 0/13 | 0/13 |
| **CoV OC43** | 13 | 0/13 | 0/13 | 0/13 | 0/13 | 0/13 | 0/13 | 0/13 |
| Inf A/H1N1 | 3 | 0/3 | 0/3 | 0/3 | 0/3 | 0/3 | 0/3 | 0/3 |
| Inf A/H3N2 | 3 | 0/3 | 0/3 | 0/3 | 0/3 | 0/3 | 0/3 | 0/3 |
| Inf B | 3 | 0/3 | 0/3 | 0/3 | 0/3 | 0/3 | 0/3 | 0/3 |
| HEV | 3 | 0/3 | 0/3 | 0/3 | 0/3 | 0/3 | 0/3 | 0/3 |
| AdV | 3 | 0/3 | 0/3 | 0/3 | 0/3 | 0/3 | 0/3 | 0/3 |
| PIV | 3 | 0/3 | 0/3 | 0/3 | 0/3 | 0/3 | 0/3 | 0/3 |
| MPV | 3 | 0/3 | 0/3 | 0/3 | 0/3 | 0/3 | 0/3 | 0/3 |
| HboV | 3 | 0/3 | 0/3 | 0/3 | 0/3 | 0/3 | 0/3 | 0/3 |
| HRV | 3 | 0/3 | 0/3 | 0/3 | 0/3 | 0/3 | 0/3 | 0/3 |
| RSV A | 3 | 0/3 | 0/3 | 0/3 | 0/3 | 0/3 | 0/3 | 0/3 |
| RSV B | 3 | 0/3 | 0/3 | 0/3 | 0/3 | 0/3 | 0/3 | 0/3 |

CoV 229E: Human coronavirus 229E, CoV NL63: Human coronavirus NL63, CoV OC43: Human coronavirus OC43, Inf A/H1N1: Influenza A type H1N1, Inf A/H3N2: Influenza A type H3N2, Inf B: Influenza B, HEV: Human enterovirus, AdV: Adenovirus, PIV: Parainfluenza virus, MPV: Human metapneumovirus, HboV: Human bocavirus, HRV: Human rhinovirus, RSV A: Respiratory syncytial virus A, RSV B: Respiratory syncytial virus B.

(MERS CoV) and severe acute respiratory syndrome coronavirus (SARS CoV) [18,19]. The genome of SARS CoV-2 consists of approximately 30,000 bases [20,21]. A phylogenetic analysis revealed that genome sequences of SARS CoV-2 from different patients were extremely similar (with 99.98% identity) and that SARS CoV-2 was closely related (with 88% identity) to two bat-derived SARS-like coronaviruses, bat-SL-CoVZC45 and bat-SL-CoVZXC21, collected in 2018 in Zhoushan, in eastern China [22].

Commercial SARS CoV-2 diagnostic RT-PCR kits detect 2–3 genes to produce a more accurate diagnosis of SARS CoV-2. Since the sensitivity of each primer set in clinical samples may be different, it is diagnosed as a positive sample when 2–3 genes are all positive, and if only one is identified, it is re-tested with another kit. Therefore, the multiplex primer set to detect two more genes is important in developing the SARS CoV-2 LAMP kit. Currently, several SARS CoV-2 LAMP primer sets were reported [23–26]. They were mostly developed with fast colorimetric detection of one or two genes suitable for on-site diagnosis [27–30]. However, it has disadvantages in not producing diagnose with multiplex testing and having to test each primer set individually. In particular, the LAMP assay has been reported to be highly susceptible to contamination [31,32], and the recently reported SARS CoV-2 RT-LAMP assay has also pointed out such a problem [33]. Therefore, if an RT-LAMP test for one clinical sample is performed with three or four LAMP primer sets (including internal control) individually, the degree of contamination may also increase. In addition, when conducting clinical tests in large quantities, the number of clinical trials more than doubles, and the advantage of a rapid diagnosis of the LAMP assay may be diluted. Therefore, the multiplex SARS CoV-2 RdRP/N/IC RT-LAMP assay developed in this study has an advantage in minimizing the contamination and enabling a mass diagnosis.

In this study, we have developed the multiplex SARS CoV-2 RT-LAMP assay, including an internal control (actin beta, IC) to detect detection the RdRP and N gene of SARS CoV-2 using strand-displaceable probes. In sensitivity test for SARS CoV-2 clinical samples, the multiplex SARS CoV-2 RdRP/N/IC RT-LAMP assay showed RdRP: 93.85%, N:94.62% and RdRP/N: 96.92% for SARS CoV-2 clinical samples (n = 130). This result is comparable to that (100%) of the commercial Allplex™ 2019-nCoV Assay (Seegene, Inc., Seoul, South Korea) and superior to that (RdRP: 92.31%, E: 93.85% and RdRP/E: 95.38%) of the PowerChek™ 2019-nCoV Real-time PCR kit (Kogenebiotech, Seoul, South Korea). Furthermore, the detection limits for the multiplex SARS CoV-2 RdRP/N/IC RT-LAMP was similar to that of the Allplex™ 2019-nCoV Assay (Seegene, Inc., Seoul, South Korea) and superior to that of the commercial PowerChek™ 2019-nCoV Real-time PCR kit (Kogenebiotech, Seoul, South Korea).

Finally, the multiplex SARS CoV-2 RdRP/N/IC RT-LAMP assay showed 100% specificity, with no cross reactivity for NP samples from patients infected with other respiratory viruses (including Coronavirus 229E, NL63 and OC43) and from uninfected healthy controls. Unfortunately, the multiplex SARS CoV-2 RdRP/N/IC RT-LAMP assay was not tested for cross reactivity against SARS CoV or other bat-derived SARS-like coronaviruses.

While the two types of RT-qPCR kit take 2 hours and 30 minutes of assay time, the multiplex SARS CoV-2 RdRP/N/IC RT-LAMP assay is very fast and produces results within 40 minutes, so if we use a 15-minute nucleic acid auto-extractor, it is possible to finish an assay within 1 hour. Therefore, if used with multi-channel isothermal equipment, such as a T16-ISO Instrument (Axxin, Australia), it will be useful for airports, ports, emergency rooms, and drive thru type SARS CoV-2 testing systems.

Here, we have developed a multiplex SARS CoV-2 RdRP/N/IC RT-LAMP assay capable of detecting RdRP, N genes and IC (actin beta, IC) in a single tube. Since the multiplex SARS CoV-2 RdRP/N/IC RT-LAMP assay takes less time (approximately 40 min), compared to the commercial Allplex™ 2019-nCoV Assay and PowerChek™ 2019-nCoV Real-time PCR kit (usually 2–3 hours), it shows promise for deployment as an on-site molecular diagnostic test.

## Supporting information

**S1 Fig. Limit of detection of the multiplex SARS CoV-2 RdRP/N/IC RT-LAMP assay for non-infected clinical samples (ranging from 1 to $10^{-3}$ copies/μL).** Numbers (1–5) indicated diluted samples/μL ($1.0$–$1.0 \times 10^{-3}$ copies/μL) and negative control (distilled water (DW) as non-template control).
(DOCX)

## Author Contributions

**Conceptualization:** Woong Sik Jang, Jung Yoon, Jeonghun Nam, Nam-Hee Ryoo, Chae Seung Lim.

**Data curation:** Woong Sik Jang, Ahran Kim, Richard Yanagihara, Nam-Hee Ryoo, Chae Seung Lim.

**Formal analysis:** Woong Sik Jang, Bo Kyeung Jung, Nam-Hee Ryoo, Chae Seung Lim.

**Funding acquisition:** Chae Seung Lim.

**Investigation:** Da Hye Lim, Minsup Lim, Sook-Won Ryu, Nam-Hee Ryoo.

**Methodology:** Da Hye Lim, Minsup Lim.

**Project administration:** Chae Seung Lim.

**Resources:** Nam-Hee Ryoo.

**Supervision:** Chae Seung Lim.

**Validation:** Nam-Hee Ryoo.

**Writing – original draft:** Woong Sik Jang.

**Writing – review & editing:** Jung Yoon, Jeonghun Nam, Richard Yanagihara, Nam-Hee Ryoo, Chae Seung Lim.

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
