## [Decision Letter · Decision Letter 0]

12 Jan 2021

PONE-D-20-37870

Development of a Multiplex Loop-Mediated Isothermal Amplification (LAMP) Assay for On-Site Diagnosis of SARS CoV-2

PLOS ONE

Dear Dr. Lim,

Thank you for submitting your manuscript to PLOS ONE. After careful consideration, we feel that it has merit but does not fully meet PLOS ONE’s publication criteria as it currently stands. Therefore, we invite you to submit a revised version of the manuscript that addresses the points raised during the review process.

In particular it is requested to increase your sample size and provide a sample size calculation for this study. Increasing the number of samples/patients and including more non-Covid cases would make this manuscript acceptable.

We look forward to receiving your revised manuscript.

Kind regards,

Henk D. F. H. Schallig, Ph.D

Academic Editor

PLOS ONE

Journal Requirements:

Reviewers' comments:

Reviewer's Responses to Questions

**Comments to the Author**

1. Is the manuscript technically sound, and do the data support the conclusions?

Reviewer #1: Yes

2. Has the statistical analysis been performed appropriately and rigorously? 

Reviewer #1: No

3. Have the authors made all data underlying the findings in their manuscript fully available?

Reviewer #1: Yes

4. Is the manuscript presented in an intelligible fashion and written in standard English?

Reviewer #1: Yes

5. Review Comments to the Author

Reviewer #1: This manuscript describes (another) development and evaluation of a LAMP assay for Covid19. There are many publications describing such a test (see list below). Consequently the authors should make clear why their test has an add on value.

Although the work is well designed and presented, a concern is that the sample size is rather limited. A sample size calculation is not provided. This must be done.

The paper would stand out amongst all other papers in this particular field if they would have assesses a much larger sample size. This is not the case. Therefore, I recommend that the authors should extend their study population before re-submitting the manuscript.

A non-template control should be included in the assay.

The resolution of figure 2 is rather low.

Loop mediated isothermal amplification (LAMP) assays as a rapid diagnostic for COVID-19.

Kashir J, Yaqinuddin A.Med Hypotheses. 2020 Aug;141:109786. doi: 10.1016/j.mehy.2020.109786. Epub 2020 Apr 25.PMID: 32361529 Free PMC article.

Loop-Mediated Isothermal Amplification (LAMP): A Rapid, Sensitive, Specific, and Cost-Effective Point-of-Care Test for Coronaviruses in the Context of COVID-19 Pandemic.

Augustine R, Hasan A, Das S, Ahmed R, Mori Y, Notomi T, Kevadiya BD, Thakor AS.Biology (Basel). 2020 Jul 22;9(8):182. doi: 10.3390/biology9080182.PMID: 32707972 Free PMC article.

A Rapid, Simple, Inexpensive, and Mobile Colorimetric Assay COVID-19-LAMP for Mass On-Site Screening of COVID-19.

Chow FW, Chan TT, Tam AR, Zhao S, Yao W, Fung J, Cheng FK, Lo GC, Chu S, Aw-Yong KL, Tang JY, Tsang CC, Luk HK, Wong AC, Li KS, Zhu L, He Z, Tam EWT, Chung TW, Wong SCY, Que TL, Fung KS, Lung DC, Wu AK, Hung IF, Woo PC, Lau SK.Int J Mol Sci. 2020 Jul 29;21(15):5380. doi: 10.3390/ijms21155380.PMID: 32751106 Free PMC article.

Rapid Detection of COVID-19 Coronavirus Using a Reverse Transcriptional Loop-Mediated Isothermal Amplification (RT-LAMP) Diagnostic Platform.

Yu L, Wu S, Hao X, Dong X, Mao L, Pelechano V, Chen WH, Yin X.Clin Chem. 2020 Jul 1;66(7):975-977. doi: 10.1093/clinchem/hvaa102.PMID: 32315390 Free PMC article.

RT-LAMP for rapid diagnosis of coronavirus SARS-CoV-2.

Huang WE, Lim B, Hsu CC, Xiong D, Wu W, Yu Y, Jia H, Wang Y, Zeng Y, Ji M, Chang H, Zhang X, Wang H, Cui Z.Microb Biotechnol. 2020 Jul;13(4):950-961. doi: 10.1111/1751-7915.13586. Epub 2020 Apr 25.PMID: 32333644 Free PMC article.

Use of the variplex™ SARS-CoV-2 RT-LAMP as a rapid molecular assay to complement RT-PCR for COVID-19 diagnosis.

Rödel J, Egerer R, Suleyman A, Sommer-Schmid B, Baier M, Henke A, Edel B, Löffler B.J Clin Virol. 2020 Nov;132:104616. doi: 10.1016/j.jcv.2020.104616. Epub 2020 Aug 31.PMID: 32891938 Free PMC article.

6. PLOS authors have the option to publish the peer review history of their article (what does this mean?). If published, this will include your full peer review and any attached files.

Reviewer #1: No

---

## [Author Response · Author response to Decision Letter 0]

17 Feb 2021

Reviewer #1: 

1. This manuscript describes (another) development and evaluation of a LAMP assay for Covid19. There are many publications describing such a test (see list below). Consequently the authors should make clear why their test has an add on value.

Loop mediated isothermal amplification (LAMP) assays as a rapid diagnostic for COVID-19. Kashir J, Yaqinuddin A.Med Hypotheses. 2020 Aug;141:109786. doi: 10.1016/j.mehy.2020.109786. Epub 2020 Apr 25.PMID: 32361529 Free PMC article.

Loop-Mediated Isothermal Amplification (LAMP): A Rapid, Sensitive, Specific, and Cost-Effective Point-of-Care Test for Coronaviruses in the Context of COVID-19 Pandemic. Augustine R, Hasan A, Das S, Ahmed R, Mori Y, Notomi T, Kevadiya BD, Thakor AS.Biology (Basel). 2020 Jul 22;9(8):182. doi: 10.3390/biology9080182.PMID: 32707972 Free PMC article.

A Rapid, Simple, Inexpensive, and Mobile Colorimetric Assay COVID-19-LAMP for Mass On-Site Screening of COVID-19. Chow FW, Chan TT, Tam AR, Zhao S, Yao W, Fung J, Cheng FK, Lo GC, Chu S, Aw-Yong KL, Tang JY, Tsang CC, Luk HK, Wong AC, Li KS, Zhu L, He Z, Tam EWT, Chung TW, Wong SCY, Que TL, Fung KS, Lung DC, Wu AK, Hung IF, Woo PC, Lau SK.Int J Mol Sci. 2020 Jul 29;21(15):5380. doi: 10.3390/ijms21155380.PMID: 32751106 Free PMC article.

Rapid Detection of COVID-19 Coronavirus Using a Reverse Transcriptional Loop-Mediated Isothermal Amplification (RT-LAMP) Diagnostic Platform. Yu L, Wu S, Hao X, Dong X, Mao L, Pelechano V, Chen WH, Yin X.Clin Chem. 2020 Jul 1;66(7):975-977. doi: 10.1093/clinchem/hvaa102.PMID: 32315390 Free PMC article.

RT-LAMP for rapid diagnosis of coronavirus SARS-CoV-2. Huang WE, Lim B, Hsu CC, Xiong D, Wu W, Yu Y, Jia H, Wang Y, Zeng Y, Ji M, Chang H, Zhang X, Wang H, Cui Z.Microb Biotechnol. 2020 Jul;13(4):950-961. doi: 10.1111/1751-7915.13586. Epub 2020 Apr 25.PMID: 32333644 Free PMC article.

Use of the variplex™ SARS-CoV-2 RT-LAMP as a rapid molecular assay to complement RT-PCR for COVID-19 diagnosis. Rödel J, Egerer R, Suleyman A, Sommer-Schmid B, Baier M, Henke A, Edel B, Löffler B.J Clin Virol. 2020 Nov;132:104616. doi: 10.1016/j.jcv.2020.104616. Epub 2020 Aug 31.PMID: 32891938 Free PMC article.

-> As you mention above, all case of SARS CoV-2 RT-LAMP assay showed good performance. However, they used one reaction mixture in one tubes for detecting one gene of SARS CoV-2. Thus, for detecting two gene of SARS CoV-2 and internal control gene in order to increase sensitivities, they need at least three reaction tubes. Furthermore, it is known that LAMP assay is easy to contaminate. So, our multiplex LAMP assay in one tube have advantages for reducing test time and risk of contamination. These contents were included in discussion following as:

“Currently, several SARS CoV-2 LAMP primer sets were reported [23-26]. They were mostly developed with fast colorimetric detection of one or two genes suitable for on-site diagnosis [27-30]. However, it has disadvantages in not producing diagnose with multiplex testing and having to test each primer set individually. In particular, the LAMP assay has been reported to be highly susceptible to contamination [31, 32], and the recently reported SARS CoV-2 RT-LAMP assay has also pointed out such a problem [33]. Therefore, if an RT-LAMP test for one clinical sample is performed with three or four LAMP primer sets (including internal control) individually, the degree of contamination may also increase. In addition, when conducting clinical tests in large quantities, the number of clinical trials more than doubles, and the advantage of a rapid diagnosis of the LAMP assay may be diluted. Therefore, the multiplex SARS CoV-2 RdRP/N/IC RT-LAMP assay developed in this study has an advantage in minimizing the contamination and enabling a mass diagnosis.” (line 263-274)

2. Although the work is well designed and presented, a concern is that the sample size is rather limited. A sample size calculation is not provided. This must be done.

-> We added sentence based on the comment following as:

“To estimate the number of samples required for clinical test of the multiplex RT-LAMP assay, the following formula was used:

n≥(〖(1.96)〗^2 p(1-p))/x^2 

where p is the suspected sensitivity, and x is the desired margin of error [15, 16]. The true-positive rate (sensitivity) was defined as the proportion of SARS-CoV-2 positive which is correctly identified by the multiplex RT-LAMP assay compared to the AllplexTM 2019-nCoV Assay (Seegene, Inc., Seoul, South Korea). We suspected the sensitivity and specificity of the multiplex RT-LAMP assay to be 95% with a desired margin of error of 0.04%. Under these conditions, the number of required samples is 114.0475 (rounded up to 115) per group. In this experiment, we have tested total 292 samples (130 positive and 162 negative).” (Line 92-101)

3. The paper would stand out amongst all other papers in this particular field if they would have assesses a much larger sample size. This is not the case. Therefore, I recommend that the authors should extend their study population before re-submitting the manuscript.

-> As your suggestion, we assess the multiplex LAMP assay with 293 clinical samples (130 SARS CoV-2 samples and 162 negative samples.

4. A non-template control should be included in the assay.

-> We have added the non-template control in Figure 1 Legend based on the comment following as:

“Numbers (1-10) indicated plasmid copy numbers/μL (1.0 × 108 - 1.0 × 100 copies/μL) and negative control (distilled water (DW) as non-template control).” (Line 474-475)

5. The resolution of figure 2 is rather low.

-> We have corrected the resolution of figure 2 based on the comment.

---

## [Decision Letter · Decision Letter 1]

19 Feb 2021

Development of a Multiplex Loop-Mediated Isothermal Amplification (LAMP) Assay for On-Site Diagnosis of SARS CoV-2

PONE-D-20-37870R1

Dear Dr. Lim,

We’re pleased to inform you that your manuscript has been judged scientifically suitable for publication and will be formally accepted for publication once it meets all outstanding technical requirements.

Kind regards,

Henk D. F. H. Schallig, Ph.D

Academic Editor

PLOS ONE

Additional Editor Comments (optional):

Reviewers' comments:

Reviewer's Responses to Questions

**Comments to the Author**

1. If the authors have adequately addressed your comments raised in a previous round of review and you feel that this manuscript is now acceptable for publication, you may indicate that here to bypass the “Comments to the Author” section, enter your conflict of interest statement in the “Confidential to Editor” section, and submit your "Accept" recommendation.

Reviewer #1: All comments have been addressed

2. Is the manuscript technically sound, and do the data support the conclusions?

Reviewer #1: Yes

3. Has the statistical analysis been performed appropriately and rigorously? 

Reviewer #1: Yes

4. Have the authors made all data underlying the findings in their manuscript fully available?

Reviewer #1: Yes

5. Is the manuscript presented in an intelligible fashion and written in standard English?

Reviewer #1: Yes

6. Review Comments to the Author

Reviewer #1: Well revised! Clear sample size estimation and it is noted what the advantage of this approach is compared to other assays

7. PLOS authors have the option to publish the peer review history of their article (what does this mean?). If published, this will include your full peer review and any attached files.

Reviewer #1: No

---

## [Editor Report · Acceptance letter]

23 Feb 2021

PONE-D-20-37870R1 

Development of a Multiplex Loop-Mediated Isothermal Amplification (LAMP) Assay for On-Site Diagnosis of SARS CoV-2 

Dear Dr. Lim:

I'm pleased to inform you that your manuscript has been deemed suitable for publication in PLOS ONE. Congratulations! Your manuscript is now with our production department. 

Kind regards, 

on behalf of

Dr. Henk D. F. H. Schallig 

Academic Editor

PLOS ONE